

# Brief communication: A fast vortex-based smearing correction for the actuator line

Alexander R Meyer Forsting[1], Georg R Pirrung[1], and Néstor Ramos-García [1]

[1]DTU Wind Energy, Technical University of Denmark, Frederiksborgvej 399, 4000 Roskilde, Denmark

**Correspondence:** Alexander R Meyer Forsting (alrf@dtu.dk)

**Abstract.** The actuator line is a lifting line representation of aerodynamic surfaces in computational fluid dynamics applications, but with non-singular forces which reduces the self-induced velocities at the line. The vortex-based correction by Meyer Forsting et al. (2019a) *et al.* recovers this missing induction and thus the intended lifting line behaviour of the actuator line. However, its computational cost exceeds that of existing tip corrections and quickly grows with blade discretization. Here we present different methods for reducing its computational cost to the level of existing corrections without jeopardising the stability or accuracy of the original method. The cost is reduced by at least 98% whereas the power is maximally affected by 0.8% with respect to the original formulation.

## 1 Introduction

The actuator line (AL) Sørensen and Shen (2002) is a lifting line (LL) representation of aerodynamic surfaces in Eulerian computational fluid dynamics (CFD) applications. It allows simulating the interaction between the atmosphere and wind farms, as it captures all the important flow features of fully resolved rotors, at a fraction of the computational cost. However transferring a LL into the CFD domain, requires dispersing the concentrated blade forces of the LL over a certain region - most commonly in form of a Gaussian projection - to avoid causing numerical instabilities. This force smearing leads to the formation of a viscous core in the released vorticity, which subsequently reduces the induced velocity at the blade (Dag, 2017; Meyer Forsting et al., 2019a; Martínez-Tossas and Meneveau, 2019). Lower induction implies larger angles-of-attack and thus increased blade forces. Especially in regions presenting large load changes, as around the root and tip of the blade, does the AL thus overestimate the forces.

Meyer Forsting et al. (2019a) - following the approach proposed by Dag (2017) - presented a correction to the AL, that combines the fast and dynamic near-wake model by Pirrung et al. (2016, 2017a, b) with a viscous core model (Lamb, 1932; Oseen, 1911) to recover the missing induction. With the correction, the AL truly functions as a LL, which was verified over the entire operational wind speed range of modern turbines as well as in yaw and for dynamic pitch steps (Meyer Forsting et al.,



2019a). The numerical stability of the correction was not challenged by any of those flow cases - not even by extreme inflow turbulence.

The only disadvantage of the new smearing correction is its computational cost. Though it is incorrect to apply conventional tip corrections to ALs - they correct actuator discs for missing discrete blades - their low cost makes them attractive. In this
paper we present different methods that reduce the computational cost of the new correction to that of existing corrections without jeopardising the stability or accuracy of the method.

## 2 Methods for increasing speed

Computing the missing induction requires re-evaluating the velocity contribution from each previously released vortex element at each times step. The velocity contribution from a single trailed vortex at some point along the blade is obtained by integrating
along the vortex line

$$\boldsymbol{u}^{\star} = \int\limits_{0}^{\infty} f_{\epsilon}\boldsymbol{\delta}\tilde{\boldsymbol{u}}\,\mathrm{d}l \tag{1}$$

Here $\boldsymbol{\delta}\tilde{\boldsymbol{u}}$ is the velocity induced by an infinitesimal element $\boldsymbol{\delta}l$ of a vortex line and $f_{\epsilon}$ represents the smearing factor, originating from the presence of a viscous core in the released vorticity. Integrating over the vortex length is equivalent to integrating over time, as at each time step an element is released. Originally, the near-wake model by Pirrung et al. (2016, 2017a, b) provides
directly the integrated velocities $\tilde{\boldsymbol{u}}$.[1] It was only broken into elements, as $f_{\epsilon}$ is a function of the perpendicular distance from the vortex to the blade element, which varies in time. As the distance changes at each time step, the velocity contribution from each vortex element also needs to be updated each time step. Hence the more vortex lines, the costlier becomes the correction.

### 2.1 Reduce wake length (orig. $\beta_{\mathbf{max}} = \pi/2$)

In the work verifying the smearing correction by Meyer Forsting et al. (2019a), the integration along the vortex lines was
20 performed until $\beta_{\max} = 2\pi$, where $\beta$ defines the rotation angle, to ensure most induction is captured. However, the near-wake model is devised to provide only the induction from the vortex lines until $\beta = \pi/2$. Considering that the vortex core effect is only active in the near-wake, $\beta_{\max}$ could equally be set to $\pi/2$, thus reducing the number of vortex elements significantly.

### 2.2 Reduce inner loops (cut loops)

The computational cost of vortex methods grows with the square of the blade elements, which could lead to escalating costs
with increasing discretization. Usually, the induction of each vortex line on each blade section needs to be determined. Yet the limited size of the viscous core allows short-cutting this procedure by considering only the blade sections closest to the vortex

---

[1]Note that the integration only covers the near-wake region, from 0 to $\pi/2$.





line. The velocity missing in AL simulations in two-dimensions is given by

$$v^\star(r,\epsilon) = \tilde{v} \overbrace{\exp(-r^2/\epsilon^2)}^{f_\epsilon} \tag{2}$$

with $r$ representing the distance from the vortex core and $\epsilon$ the force smearing length scale. To determine the size of the vortex core $r_{\max}$, the ratio between cut and fully resolved vortex core is computed

$$5 \quad I = \frac{\int_0^{r_{\max}} v^\star(r,\epsilon)\,\mathrm{d}r}{\int_0^\infty v^\star(r,\epsilon)\,\mathrm{d}r} \tag{3}$$

Different ratios were tested, however $I = 0.99$ - corresponding to $r_{\max} = 1.83\epsilon$ - provides a beneficial balance between accuracy and speed.

### 2.3 Constant smearing factor, $f_\epsilon$ (fixed $x_\perp$)

A more radical approach than just reducing the wake length, as described in Section 2.1, is fixing the perpendicular distance
10 between the vortex and blade element and thus the smearing factor. In the three-dimensional formulation the smearing factor is given by (Meyer Forsting et al., 2019a)

$$f_\epsilon = \exp\left(-\frac{|x_\perp(r,\beta,h,\phi)|^2}{\epsilon^2}\right) \tag{4}$$

with the perpendicular distance

$$x_\perp = r\cos\phi \begin{pmatrix} \tan\phi\,(\beta\cos\beta - \sin\beta) \\ -\tan\phi\,(-1 + h/r + \cos\beta + \beta\sin\beta) \\ -1 + (1 - h/r)\cos\beta \end{pmatrix} \tag{5}$$

15 The greatest simplification is achieved by setting $\beta = 0$, such that $x_\perp$ becomes the distance between vortex trailing point and blade section, which is a geometric constant for rigid blades.

$$|x_\perp(\beta = 0)| = h \tag{6}$$

The smearing factor no longer needs to be updated for all vortex elements at each time step and the velocity correction in Eq. (1) simply becomes:

$$20 \quad u^\star = f_\epsilon \tilde{u} \tag{7}$$

where $\tilde{u}$ is directly determined by the near-wake model. In this paper this method is run in conjunction with the previous - cutting loops - approach.





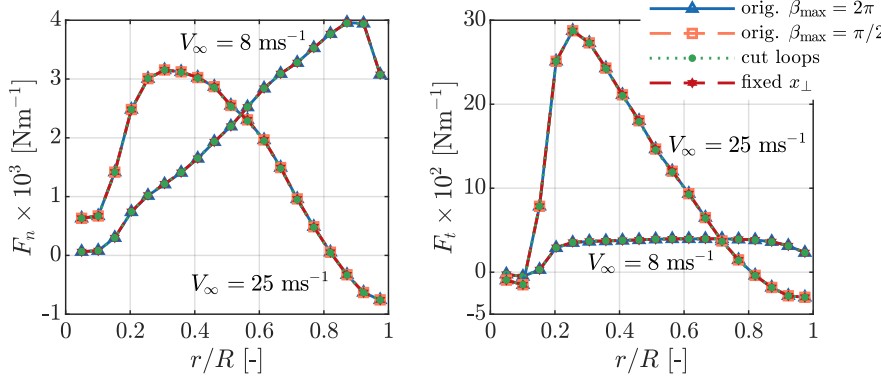

**Figure 1.** Normal and tangential forces on the NREL 5MW blades at 8 and 25 $\text{ms}^{-1}$ predicted by AL simulations (blades discretized by 19 sections) with smearing correction and different computational speed-up methods. The reference is the original formulation (orig.) with $\beta_{\max} = 2\pi$.

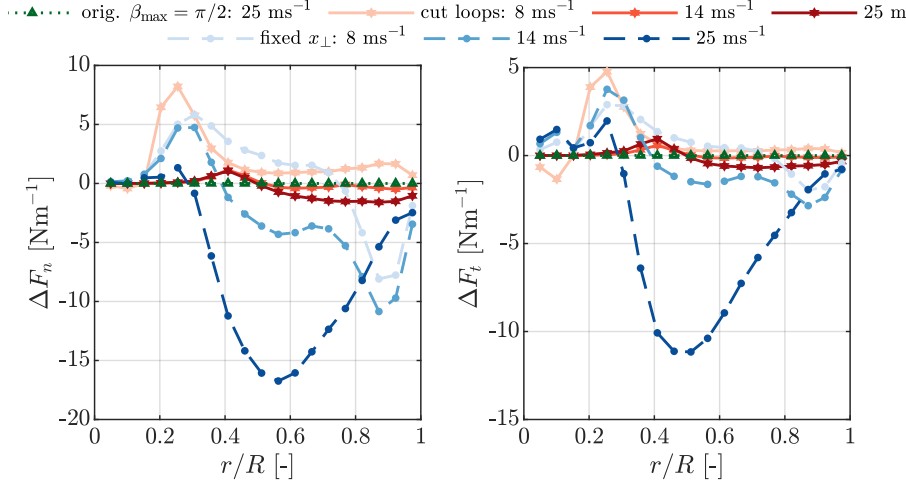

**Figure 2.** Difference in normal and tangential forces over the NREL 5MW blades at 8, 14 and 25 $\text{ms}^{-1}$ predicted by AL simulations (blades discretized by 19 sections) with different smearing correction speed-up methods with respect to simulations without speed-up.

## 3 Results

This section compares the influence of the different speed-up methods presented in Section 2 with the original results of Meyer Forsting et al. (2019a). All results are obtained with exactly the same computational setup as presented in Section 3 of the same paper. The AL models the NREL 5MW(Jonkerman et al., 2009) under uniform inflow. For the inflow wind speed specific turbine parameters refer to Table 2 in Meyer Forsting et al. (2019a).





Figure 1 compares the force distributions for the NREL 5MW at two wind speeds obtained with the speed-up methods presented in Section 2 to those obtained with the original model. The influence of reducing the wake length is only shown for a wind speed of $25\,\mathrm{ms^{-1}}$, but is similar at lower wind speeds. With increasing wind speed, the peak force moves clearly from the tip to the root whilst the smearing correction ensures the smooth behaviour towards the blade ends. From pure visual inspection

there is no change in the forces when applying any of the speed-up options. To highlight their impact, only the change in the force distributions with respect to the unmodified model is shown in Fig. 2 - here additionally the results for a wind speed of $14\,\mathrm{ms^{-1}}$ are presented. Reducing the wake length has a negligible effect on the forces as does reducing the inner loops, except close to the root. Fixing the smearing factor additionally to cutting the loops has the largest influence, however even at $25\,\mathrm{ms^{-1}}$ the deviation does not exceed $17\,\mathrm{Nm^{-1}}$. With respect to the local force the difference remains below 1%.

A full result overview - the impact of the speed-up methods on thrust and power as well as their influence on the computational cost per blade - is given in Table 1. Results are shown for rotors discretized by 9 and 19 blade sections. Firstly, the greatest change in thrust or power across all methods occurs when fixing the smearing constant, yet never by more than 0.8% and only at the highest wind speed. The positive influence of cutting the inner loops on performance grows with increasing resolution. However, the largest reduction in the computational cost arrives from limiting the wake length and ultimately fixing

the smearing factor. With the latter approach the longest of all smearing correction iterations lasted $8 \times 10^{-4}$ s.





**Table 1.** An overview of the influence of the computational speed-up methods on thrust, power and computational cost per blade for two different blade discretizations - 9 and 19 blade sections. Only for the original model are the nominal values shown, otherwise the relative change to the original is given in percent.

| $N_s$ | | $V_\infty$ [ms$^{-1}$] | orig. $\beta_{max} = 2\pi$ | orig. $\beta_{max} = \pi/2$ [%] | cut loops [%] | fixed $x_\perp$ [%] |
|---|---|---|---|---|---|---|
| 9 | Thrust | 8 | 406 kN | - | $1.96 \times 10^{-2}$ | $-3.83 \times 10^{-2}$ |
| | | 14 | 466 kN | - | $1.35 \times 10^{-2}$ | $-1.79 \times 10^{-1}$ |
| | | 25 | 286 kN | $4.43 \times 10^{-3}$ | $-2.06 \times 10^{-2}$ | $-5.65 \times 10^{-1}$ |
| | Power | 8 | 2.11 MW | - | $4.37 \times 10^{-2}$ | $-1.37 \times 10^{-1}$ |
| | | 14 | 5.43 MW | - | $1.88 \times 10^{-2}$ | $-2.75 \times 10^{-1}$ |
| | | 25 | 5.47 MW | $6.78 \times 10^{-3}$ | $-2.28 \times 10^{-2}$ | $-7.95 \times 10^{-1}$ |
| | Cost | 8 | $8.68 \times 10^{-3}$ s | - | $-43.5$ | $-99.3$ |
| | | 14 | $9.91 \times 10^{-3}$ s | - | $-44.9$ | $-98.3$ |
| | | 25 | $1.04 \times 10^{-2}$ s | $-83.4$ | $-45.9$ | $-99.4$ |
| 19 | Thrust | 8 | 394 kN | - | $9.13 \times 10^{-2}$ | $2.81 \times 10^{-2}$ |
| | | 14 | 456 kN | - | $-7.40 \times 10^{-4}$ | $-9.81 \times 10^{-2}$ |
| | | 25 | 277 kN | $8.73 \times 10^{-5}$ | $-3.30 \times 10^{-2}$ | $-4.75 \times 10^{-1}$ |
| | Power | 8 | 2.00 MW | - | $1.71 \times 10^{-1}$ | $1.43 \times 10^{-2}$ |
| | | 14 | 5.28 MW | - | $-1.80 \times 10^{-3}$ | $-1.56 \times 10^{-1}$ |
| | | 25 | 5.29 MW | $1.32 \times 10^{-4}$ | $-4.36 \times 10^{-2}$ | $-6.83 \times 10^{-1}$ |
| | Cost | 8 | $4.35 \times 10^{-2}$ s | - | $-63.3$ | $-99.0$ |
| | | 14 | $4.45 \times 10^{-2}$ s | - | $-63.7$ | $-98.2$ |
| | | 25 | $4.37 \times 10^{-2}$ s | $-82.9$ | $-62.6$ | $-99.0$ |

## 4 Conclusions

The smearing correction by Meyer Forsting et al. (2019a) recovered the lifting line behaviour of the actuator line, however at a larger computational cost than existing actuator disc tip corrections. This paper presents different methods for reducing the cost of the smearing correction to those levels. The number of wake elements manifests itself as the key cost driver. Reducing the
5  wake length therefore significantly reduces the computational cost without negatively impacting the blade forces. The greatest speed-up arrives from avoiding recomputing the contributions from each element at each time step altogether, leading to a fall in the cost of at least 98%. This is accompanied by changes in thrust and power of maximally 0.8% and 0.7%, respectively. Still, with respect to the great gain in performance this is acceptable and lies well within CFD simulation uncertainty. Furthermore, the new, faster method avoids any form of bookkeeping, greatly simplifying the implementation of the smearing correction.
10  This faster and simpler version of the smearing correction is openly available (Meyer Forsting et al., 2019b).



*Code availability.* All data are available on request. Commercial and research licenses for EllipSys3D can be purchased from DTU. The source code of the fast smearing correction is openly available (Meyer Forsting et al., 2019b).

*Competing interests.* The authors declare no conflict of interest.

*Acknowledgements.* We would like to acknowledge DTU Wind Energy's internal project "Virtual Atmosphere" for partially funding this
5   research.



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
