# Peer review of "Brief communication: A fast vortex-based smearing correction for the actuator line"

_Wind Energy Science, 2019_

## Referee Comment (RC1) · David Wood (Referee) · 6 Dec 2019

This note is an update on the authors' previous work on actuator line modeling (ALM) in which they demonstrate that substantial computational cost savings are possible in the model at only marginal degradation in accuracy. ALM is an important part of wind energy research and this contribution is an important one. My only slightly negative comment is that the symbols and lines in the figures make it difficult to determine what is plotted. I suggest a better use of open symbols, symbols without lines, and the reverse.

---

## Referee Comment (RC2) · Claudio Balzani (Referee) · 13 Jan 2020

In the paper, the authors study the impact of different techniques for the reduction of computational costs of an actuator line model on the aerodynamic force distribution along wind turbine rotor blades. The actuator line model was previously published by the authors, cf. reference Meyer Forsting et al., 2019a in the manuscript.

The paper is well written and of generally good quality. The paper is very short, matching the requirements for a brief communication. There are no unnecessary repetitions compared with the original publication, which is good. The results reveal a negligible effect on the aerodynamic force distribution while substantially reducing the computational costs. Hence, the add-on to the original formulation is of very high interest for

the wind energy research community for further investigations of the aerodynamic and aeroelastic performance of wind turbines.

There are only two suggestions to the authors: i) The authors may additionally evaluate wind turbine models that are closer to the current state of the art considering slenderness and dynamics of the blades. Such a study could be part of a follow-up publication and does not necessarily need to be implemented in the present manuscript. The authors should revise the legend of Fig. 2, as it is hard to understand what is represented by the different curves.

---

## Author Comment (AC1) · 27 Jan 2020

Dear Reviewers,

Thank you for your comments. Please see the supplement file for our response.

A Meyer Forsting, G. Pirrung and N. Ramos-García

Please also note the supplement to this comment:
https://www.wind-energ-sci-discuss.net/wes-2019-67/wes-2019-67-AC1-
supplement.pdf

---

## Author Response (AR1)

January 27, 2020

alrf

**Dear Reviewers**

First of all we would like to thank you for your very positive and constructive comments. We have tried to incorporate your suggestions and have additionally added four minor sentences throughout the paper to stress that our method is a dynamic correction. This addition does not alter any of the results or conclusions. Please find below our responses (in black) to your comments (in blue) and at the end of this letter a marked-up version showing all changes throughout the paper. We hope that you will accept the revised manuscript for publication.

Yours sincerely

AR. Meyer Forsting, G. Pirrung and N. Ramos-García

Frederiksborgvej 399 Building 125 DK-4000 Roskilde Tel +45 45 25 11 99 Dir +45 93 51 11 75

**Response to David Wood**

This note is an update on the authors' previous work on actuator line modeling (ALM) in which they demonstrate that substantial computational cost savings are possible in the model at only marginal degradation in accuracy. ALM is an important part of wind energy research and this contribution is an important one. My only slightly negative comment is that the symbols and lines in the figures make it difficult to determine what is plotted. I suggest a better use of open symbols, symbols without lines, and the reverse.

Thanks for those positive comments and we fully agree that the figures had to be improved. We have changed both figures and tried very many different combinations of marker and line styles. We hope that it is now easier to distinguish the results.

**Response to Claudio Balzani**

In the paper, the authors study the impact of different techniques for the reduction of computational costs of an actuator line model on the aerodynamic force distribution along wind turbine rotor blades. The actuator line model was previously published by the authors, cf. reference Meyer Forsting et al., 2019a in the manuscript. The paper is well written and of generally good quality. The paper is very short, matching the requirements for a brief communication. There are no unnecessary repetitions compared with the original publication, which is good. The results reveal a negligible effect on the aerodynamic force distribution while substantially reducing the computational costs. Hence, the add-on to the original formulation is of very high interest for the wind energy research community for further investigations of the aerodynamic and aeroelastic performance of wind turbines. There are only two suggestions to the authors: i) The authors may additionally evaluate wind turbine models that are closer to the current state of the art considering slenderness and dynamics of the blades. Such a study could be part of a follow-up publication and does not necessarily need to be implemented in the present manuscript. The authors should revise the legend of Fig. 2, as it is hard to understand what is represented by the different curves.

Thanks for your positive feedback. We a are pleased that this brief communication is seen as a valuable extension to our original correction. We currently are involved in different projects in which we compare our results for more modern wind turbine rotors to other models of varying fidelity, which should be part of coming publications. Those results are for solid blades, yet fully aeroelastically coupled simulations are for sure planned, as then our model has many additional benefits over the existing non-dynamic corrections. The figures were really not the best and have been thoroughly revised. Line and marker styles were updated and also the legends made more descriptive. We hope it is now clearer what is shown.

**Brief communication:** A fast vortex-based smearing correction for the actuator line**

Alexander R Meyer Forsting1, Georg R Pirrung1, and Néstor Ramos-García1 1DTU Wind Energy, Technical University of Denmark, Frederiksborgvej 399, 4000 Roskilde, Denmark **Correspondence:** Alexander R Meyer Forsting (alrf@dtu.dk)

[revised manuscript text omitted]

line. The velocity missing in AL simulations in two-dimensions is given by

$$v^{\star}(r,\epsilon) = \tilde{v} \underbrace{\exp(-r^2/\epsilon^2)}^{J_{\epsilon}}$$
(2)

with r representing the distance from the vortex core and  $\epsilon$  the force smearing length scale. To determine the size of the vortex core  $r_{\text{max}}$ , the ratio between cut and fully resolved vortex core is computed

5
$$I = \frac{\int_0^{r_{\max}} v^*(r,\epsilon) \,\mathrm{d}r}{\int_0^{\infty} v^*(r,\epsilon) \,\mathrm{d}r}$$
(3)

Different ratios were tested, however I = 0.99 - corresponding to  $r_{max} = 1.83\epsilon$  - provides a beneficial balance between accuracy and speed.

**2.3** Constant smearing factor, $f_{\epsilon}$ (fixed $x_{\perp}$ )**

A more radical approach than just reducing the wake length, as described in Section 2.1, is fixing the perpendicular distance
between the vortex and blade element and thus the smearing factor. In the three-dimensional formulation the smearing factor is given by (Meyer Forsting et al., 2019a)

$$f_{\epsilon} = \exp\left(-\frac{|\boldsymbol{x}_{\perp}(r,\beta,h,\phi)|^2}{\epsilon^2}\right)$$
(4)

with the perpendicular distance

$$\boldsymbol{x}_{\perp} = r \cos \phi \begin{pmatrix} \tan \phi \left(\beta \cos \beta - \sin \beta\right) \\ -\tan \phi \left(-1 + h/r + \cos \beta + \beta \sin \beta\right) \\ -1 + \left(1 - h/r\right) \cos \beta \end{pmatrix}$$
(5)

15 The greatest simplification is achieved by setting  $\beta = 0$ , such that  $x_{\perp}$  becomes the distance between vortex trailing point and blade section, which is a geometric constant for rigid blades.

$$|\boldsymbol{x}_{\perp}(\boldsymbol{\beta}=\boldsymbol{0})| = h \tag{6}$$

The smearing factor no longer needs to be updated for all vortex elements at each time step and the velocity correction in Eq. (1) simply becomes:

$$\quad u^{\star} = f_{\epsilon} \tilde{u} \tag{7}$$

where  $\tilde{u}$  is directly determined by the near-wake model. Thus it is very computationally efficient and does not require saving and integrating the induced velocities from discretized vortex arcs. At each time step and for each blade section, the influence from each previously trailed vortex arc can be simply updated by multiplying with an exponential decay factor and adding the influence of the newly trailed element (Pirrung et al., 2016). In this paper this method is run in conjunction with the previous – cutting loops – cutting loops approach.

Updated line/marker style and improved legend

Figure 1. Normal and tangential forces on the NREL 5MW blades at 8 and 25 ms-1 predicted by AL simulations (blades discretized by 19 sections) with smearing correction and different computational speed-up methods. The reference is the original formulation (orig.) with  $\beta_{\text{max}} = 2\pi$ .